# Scalable, Micro-Neutralization Assay for Assessment of SARS-CoV-2 (COVID-19) Virus-Neutralizing Antibodies in Human Clinical Samples

**DOI:** 10.3390/v13050893

**Published:** 2021-05-12

**Authors:** Richard S. Bennett, Elena N. Postnikova, Janie Liang, Robin Gross, Steven Mazur, Saurabh Dixit, Gregory Kocher, Shuiqing Yu, Shalamar Georgia-Clark, Dawn Gerhardt, Yingyun Cai, Lindsay Marron, Vladimir V. Lukin, Michael R. Holbrook

**Affiliations:** 1Integrated Research Facility at Fort Detrick, National Institute of Allergy and Infectious Diseases, National Institutes of Health, Frederick, MD 21702, USA; richard.bennett@nih.gov (R.S.B.); elena.postnikova2@nih.gov (E.N.P.); janie.liang@nih.gov (J.L.); robin.gross@nih.gov (R.G.); steven.mazur@nih.gov (S.M.); saurabh.dixit@nih.gov (S.D.); gregory.kocher@nih.gov (G.K.); shuiqing.yu@nih.gov (S.Y.); shalamar.georgia@nih.gov (S.G.-C.); dawn.gerhardt@nih.gov (D.G.); caiyingyun@yahoo.com (Y.C.); lindsay.marron@nih.gov (L.M.); 2A.T Kearney, Inc., Chicago, IL 60606, USA; Vladimir.Lukin@kearney.com

**Keywords:** SARS-CoV, SARS-CoV-2, coronavirus, COVID, COVID-19, neutralization, antibodies, diagnosis

## Abstract

As the severe acute respiratory syndrome coronavirus 2 (SARS-CoV-2) pandemic expanded, it was clear that effective testing for the presence of neutralizing antibodies in the blood of convalescent patients would be critical for development of plasma-based therapeutic approaches. To address the need for a high-quality neutralization assay against SARS-CoV-2, a previously established fluorescence reduction neutralization assay (FRNA) against Middle East respiratory syndrome coronavirus (MERS-CoV) was modified and optimized. The SARS-CoV-2 FRNA provides a quantitative assessment of a large number of infected cells through use of a high-content imaging system. Because of this approach, and the fact that it does not involve subjective interpretation, this assay is more efficient and more accurate than other neutralization assays. In addition, the ability to set robust acceptance criteria for individual plates and specific test wells provided further rigor to this assay. Such agile adaptability avails use with multiple virus variants. By February 2021, the SARS-CoV-2 FRNA had been used to screen over 5000 samples, including acute and convalescent plasma or serum samples and therapeutic antibody treatments, for SARS-CoV-2 neutralizing titers.

## 1. Introduction

Severe acute respiratory syndrome coronavirus 2 (SARS-CoV-2), a betacoronavirus with a positive-strand RNA genome, was identified as a novel pathogen and causative agent of coronavirus disease (COVID-19) in humans. The high transmissibility of the virus has led to a pandemic with over 110 million documented infections worldwide and over 2.4 million deaths (as of 22 February 2021).

While initially thought to cause a primarily respiratory disease, time has shown that SARS-CoV-2 infection can cause multiple clinical manifestations, including severe respiratory disease, cardiovascular disease [1], and neurological disease [2,3]. In addition, convalescence in some people can be complicated by long-term sequelae that can be quite severe [2,3,4]. Early on, individuals with pre-existing comorbidities were identified as being more severely affected by SARS-CoV-2 infection [5,6], and there are indications that age, gender, race, and other genetic factors play a role in disease severity and clinical outcome [7,8].

During acute disease, there appears to be a rapid antibody class switch from immunoglobulin M (IgM) to IgG and IgA [9,10], although a slow class switch may be predictive of patient prognosis and associated with patients requiring hospitalization [11]. The antibody isotype is important in controlling the disease, and so is the target viral protein. In fact, more robust and prolonged antibody responses to the viral nucleoprotein (N) were associated with more severe disease [11].

The presence of anti-SARS-CoV-2 antibodies in blood is presumed to be a good measure of protective immunity for a vaccine candidate. Hence, methods to reliably, sensitively, and rapidly detect SARS-CoV-2 neutralizing antibodies are needed for pre-clinical vaccine studies and clinical trials. Further, quantifying potent neutralizing antibodies from recovered COVID-19 patients may be useful in identifying potential donors for passive immunization and hyper-immunoglobulin therapeutic applications. The U.S. Food and Drug Administration (FDA) initially approved an expanded access program (EAP) [12] for the treatment of COVID-19 using plasma from individuals with a neutralization titer of 1:160 or higher. This program led to the treatment of over 94,000 patients at participating provider locations across the U.S. Based on data from the EAP [13], in August 2020, the FDA issued an emergency use authorization (EUA) to allow therapeutic plasma treatment of COVID-19 patients outside the context of clinical trials [14]. Additional efforts to develop therapeutic monoclonal antibodies have led to approval of EUAs for antibody cocktails developed by Regeneron [15] and Lilly [16].

Here, we describe the development of a semi-high-throughput SARS-CoV-2 neutralization assay that takes advantage of the capabilities of a high-content imaging system to quantify the number of infected cells in individual wells. This assay is devoid of subjective interpretation and more precise than most other wild-type virus neutralization assays. In addition, the assay has been quickly adapted for use with multiple virus variants.

## 2. Materials and Methods

### 2.1. Virus and Cells

The 2019-nCoV/USA-WA1-A12/2020 human isolate of severe acute respiratory syndrome coronavirus 2 (SARS-CoV-2) (*Nidovirales: Coronaviridae: Sarbecovirus*) was provided by the U.S. Centers for Disease Control and Prevention (CDC; Atlanta, GA, USA). The virus was propagated at the Integrated Research Facility–Frederick in high containment (biosafety level 3 [BSL-3]) by inoculating Vero cells, acquired from the American Type Culture Collection (ATCC #CCL-81; Manassas, VA, USA). The infected cells were incubated for 72 h in Dulbecco’s Modified Eagle Medium with L-glutamine (DMEM; Lonza, Walkersville, MD, USA) containing 2% heat-inactivated fetal bovine serum (FBS; SAFC Biosciences, Lenexa, KS, USA) in a humidified atmosphere at 37 °C with 5% carbon dioxide (CO_2_). The resulting master stock (IRF0394) was quantified by plaque assay using Vero E6 cells (ATCC #CRL-1586) with a 2.5% Avicel overlay and stained after 48 h with a 0.2% crystal violet stain. Working stocks (IRF0395 and IRF0399) were prepared using multiplicity of infections (MOI) of 0.01, harvested after 48 h, and quantified by plaque assay. Virus stocks were sequenced and found to be identical to the published sequence (GenBank #MT020880) for this isolate.

For the neutralization assay, Vero E6 cells (BEI #NR596; Manassas, VA, USA) were plated at a density of 3 × 10^4^ cells per well in 96-well plates and incubated overnight at 37 °C with 5% CO_2_ so they were approximately 80–90% confluent on the day of infection.

### 2.2. Sample Dilution

The fluorescence reduction neutralization assay (FRNA) method for measuring neutralizing antibodies was originally developed for Ebola virus and Middle East respiratory syndrome coronavirus (MERS-CoV) [17,18]. Two changes were made to adapt it to measure SARS-CoV-2-specific antibodies. First, cell culture dilution media without calcium was used to reduce sample coagulation, particularly when screening plasma samples. Second, statistical evaluations to allow masking of outlier data were developed.

Different positive controls have been used in this assay depending upon the material being tested and its application. Most assays were run using a SARS-CoV-2 antiserum (SAB Biotherapeutics, Sioux Falls, SD, USA) as a positive neutralizing control, while others used an in-house hyperimmune anti-SARS-CoV-2 immunoglobulin. Virus and antibody-free cell culture medium served as a negative control. During assay development, it was found that there was little difference between using heat-inactivated or untreated serum or plasma samples. Subsequently, in an effort to decrease turnaround time, samples were not heat-inactivated, and no complement was added. For standard plasma screening, samples (positive control antibody and test article) were diluted through a six-step two-fold serial dilution (1:40–1:1280) in serum-free DMEM (Gibco #21068028, Gaithersburg, MD, USA) using a 96-well plate format (Figure 1A). For studies with higher titer monoclonal antibodies, requiring precise calculation of a 50% neutralizing titer (NT_50_), a 12-step two-fold serial dilution (e.g., 20–40,960) was used (Figure 1B). SARS-CoV-2 was diluted in serum-free DMEM to MOI 0.5 (e.g., 15,000 PFU per 30,000 cells). The diluted virus and diluted test samples were mixed 1:1 (vol/vol) and incubated at 37 °C in a humidified 5% CO_2_ atmosphere for 1 h to allow anti-SARS-CoV-2 antibodies to bind the virus. The virus/sample mixtures were then transferred to the wells of a 96-well plate (Greiner Bio-One #655948, Monroe, NC, USA) containing Vero E6 cells and incubated at 37 °C and 5% CO_2_ for 24 h. For screening activities, samples were tested in duplicate on duplicate plates to allow for four replicates per tested sample. For full plate dilutions, samples were tested as a single replicate per plate on four different plates. A summary of assay set-up in provided in Table 1.

All plasma and serum samples used for developing this assay were deidentified donor samples.

### 2.3. Cell Staining

After 24 h of incubation, cells were fixed by adding 20% neutral-buffered formalin (NBF) (Thermo Scientific #23-751-800, Kalamazoo, MI, USA) directly to the media for 30 min at room temperature. Plates were stored approximately 24h in 10% NBF at 4 °C in accordance with the facility’s safety protocols and then removed from the containment laboratory. Following removal from the laboratory, the NBF was decanted, and the cells were washed twice with 1X phosphate-buffered saline (PBS) diluted with purified water from a 10X stock solution (Thermo Fisher Scientific #BP3994, Waltham, MA, USA). The cells were then permeabilized with 0.25% Triton buffer in 1X PBS (Fisher Scientific #PR-H5142) for 5 min at room temperature. Cells were then washed three times with 1X PBS prior to blocking with 3% bovine serum albumin (BSA; Sigma #A7906, Saint Louis, MO, USA) in 1X PBS. Cells were stained with an anti-SARS antibody (1:8000 dilution, SARS-CoV/SARS-CoV-2 nucleocapsid antibody, rabbit monoclonal antibody; Sino Biological #40143-R001, Wayne, PA, USA), diluted to 0.125 μg/mL in 3% BSA/PBS blocking solution for 1 h at room temperature. The cells were again washed three times with PBS and then stained with an Alexa Fluor 594-conjugated goat anti-rabbit IgG (H + L) highly cross-adsorbed secondary antibody (Thermo Fisher Scientific #A11037), diluted in BSA/PBS for 1 h in the dark at room temperature, and counterstained with Hoechst 33342 nuclear stain (Thermo Fisher Scientific #H3570). The Operetta CLS High-Content Analysis System (Perkin Elmer, Waltham, MA, USA) was used to count the number of virus-infected and non-infected cells in wells containing samples/replicates—specifically those in the four internal (not near well walls) fields with a minimum of 1000 cells per field.

### 2.4. Calculation of Standard NT_50_ Values

For screening relatively low titer samples, a standard dilution scheme of four replicates for each sample were spread across two plates (Figure 1A). The NT_50_ value was determined individually for each set of two plates based on the virus-positive control wells. To determine the 50% infection rate for a plate, the average of 12 observations of positive control wells was multiplied by 0.50. The output of each step-wise dilution was the average of the four replicates across two plates. The sample dilution output was compared against the calculated NT_50_ cutoff value for the duplicate plates, and the highest dilution to achieve an infection of ≤50% was considered the NT_50_ titer for the sample. Results are reported as the reciprocal dilution.

### 2.5. Calculation of NT_50_ Values by Regression Analysis

For larger dilution series, particularly with known high titer material such as monoclonal antibodies, a more precise calculation was required. Samples were diluted across the plate (Figure 1B) with one replicate per plate in four plates. The fluorescence signal was plotted against the log value of the antibody dilution. A four-parameter logistical analysis was performed on the full dilution series using Prism (GraphPad Software, San Diego, CA, USA). The regression was performed using all four replicates per dilution, and the precise titer was calculated from the regression curve.

### 2.6. ELISAs

Enzyme-linked immunosorbant assay (ELISA) kits from Euroimmun US, Inc., Mountain Lakes, NJ, USA (#EI 2606-9601 G) and COVID-SeroIndex from USA R&D Systems, Inc., Minneapolis, MN, USA (#DSR200) were used to test anti-SARS-CoV-2 antibody positivity of serum used in FRNA_50_. The ELISA was performed according to manufacturer instructions.

Briefly, the EUROIMMUN anti-SARS-CoV-2 assay provided a semi-quantitative in vitro determination of human antibodies of immunoglobulin classes IgG. Each kit contained microplate coated with S1 domain of spike protein of SARS-CoV-2. In the first reaction step, diluted patient samples were incubated in the wells. In the case of positive samples, S1-specific antibodies would bind to the viral antigens. To detect the bound antibodies, a second incubation was carried out using an enzyme-labelled antihuman IgG (peroxidase conjugate) catalyzing a color reaction. Results were evaluated semi-quantitatively by calculation of a ratio of the extinction of the control or patient sample over the extinction of the calibrator. This ratio was interpreted as follows: <0.8 negative; ≥0.8 to <1.0 borderline; ≥1.1 positive. Borderline results were considered positive for analysis [19].

The COVID-SeroIndex Kit from R&D utilizes a recombinant receptor binding domain (RBD) of the SARS-CoV-2 spike protein. Antigen was pre-coated onto a 96-well microplate in phase 1. When the test sample was added, antibodies specific for the SARS-CoV-2 RBD antigen bound the antigen and were retained in the well. After washing, an enzyme-linked monoclonal antibody specific for human IgG was added to the wells. Following a wash to remove any unbound enzyme-linked antibody, a substrate was added to the wells, and color developed in proportion to the amount of IgG antibodies in the sample bound to the SARS-CoV-2 RBD antigen. The color development was stopped, and the intensity of the color was measured. The cut off index was calculated as the ratio of corrected OD of the sample and corrected OD of positive control. The cut off index was interpreted as follows: <0.7 negative; ≥0.7 is positive [20].

## 3. Results

### 3.1. Immunofluorescence Staining

Development of immunofluorescence assays for Ebola virus and MERS-CoV [18] established fundamental protocols. Early in the COVID-19 pandemic (January 2020), a primary concern was identifying a SARS-CoV-2-specific antibody that reacted with this novel virus. Previous work with MERS-CoV had shown that an N-protein-specific antibody was highly reactive and very specific. Subsequently, an antibody specific for the SARS-CoV N protein proved to cross-react with SARS-CoV-2 with no marked evidence of nonspecific binding to cellular proteins (Figure 2).

### 3.2. Initial Testing of Human Plasma

Initial sample screening protocols were developed to qualify donor samples for clinical trials supported by the National Institute of Allergy and Infectious Diseases (Clinical Trials NCT04344977, NCT04546581). A donor was eligible if their neutralization titer was ≥1:80. To identify eligible samples, a two-fold serial dilution series was devised to test for titers ranging 1:40–1:1280. In addition, initial testing evaluated the utility of heat-inactivating the test article and a possible preference for serum or plasma as a preferred matrix for screening activities. This testing found that heat-inactivation did not significantly affect testing, and results between serum and plasma were similar. Thus, in order to retain consistency with the samples that would ultimately be collected from donors and to decrease processing time, subsequent screening was performed on plasma that was not heat-inactivated.

Further testing of plasma, serum, purified IgG, monoclonal antibodies (mAb) and mAb cocktails, polyclonal antibodies. and nanobodies has also been completed using this assay. As appropriate for some samples, the assay employed an alternate dilution series (e.g., 1:20–1:40,960). The fundamental aspects of the assay were unchanged regardless of the dilution series. By February 2021, over 5000 samples had been screened using this assay.

### 3.3. Outliers in the Virus Control and Cell Control Observations

Early in development of this assay, outlier wells in replicate samples of individual dilutions confounded data analysis. To address concerns with outlier data and to improve plate acceptance criteria, statistical evaluations were used to identify outliers for exclusion from data analyses and to define the number of outliers that could be excluded while retaining the viability of the analysis.

The virus positive control consisted of a total of 12 observations (six observations on each of two plates). The virus positive control was used to estimate the NT_50_, the threshold value used for detecting efficacy of a sample under specific dilution. The cell-only control also consisted of 12 observations, with six observations per plate; it was used to estimate NT_100_, another threshold value. These FRNA criteria determined whether sample readings for a given dilution were accepted or rejected.

The results of 15 experimental runs (30 plates with 2280 observations in total) were analyzed. Box plots were produced from 180 virus control and 180 cell control observations (Figure 3 and Figure 4).

Placing control data side-by-side helped to visualize extreme results. For, example, in the virus control box plot results (Figure 3), the measurements of Plate “052020_10-20, Plate 1” were concentrated close to 100%. Compared to other observations, these results may have indicated that this plate was “over-exposed” with the virus cells, and the true measurement could not have been properly estimated. At the same time, extreme values around 0% were observed at plates “052020_17-33_17-34, Plate 2”, “052120_4-7_4-8, Plate 1”, and “052120_4-7_4-8, Plate 2”. The question was whether these observations should be discarded or corrected.

The histogram of the virus control observations (Figure 5) showed concentrations above 60% (mean = 66.8%). The tails of the normal distribution curve were outside of the fixed interval of 0–100%. To estimate outliers, data were fitted to the beta distribution, which was a more appropriate choice for values in a finite interval. To estimate beta distribution parameters, the method of moments was used, which allowed the calculation of alpha and beta distributional parameters using the sample mean and standard deviation. Equations for alpha and beta using method of moments: $\hat{\alpha }=\mu \left(\frac{\mu \left(1-\mu \right)}{s^{2}}-1\right)$; $\hat{\beta }=\left(1-\mu \right)\left(\frac{\mu \left(1-\mu \right)}{s^{2}}-1\right)$, where *μ* is the sample mean and *s^2^* is the sample variance (Owen, 2008). For cases in which either $\hat{\alpha }$ or β is less than one, it is preferred to use an alternative maximum-likelihood estimation method of estimating these parameters. In those cases, we used the scipy.stats.beta.fit() command, a beta-fitting function in the statistical library within Python 3.6 2 [21,22]. To determine the boundary of acceptable values, 5% and 95% intervals of the beta (2.3, 1.1) were used. Increasing the acceptable region beyond the cutoffs would have increased the chance of accepting experimental errors and bias the sample statistics. Removing virus control observations below 24.7% and above 94.6% (outliers) shifted the sample mean from 66.8% to 67.8%, which improved the precision of detecting the proper dilution ratio (Figure 6).

The histogram of the cell control observations (Figure 7) showed data that were heavily skewed toward 0%. Thus, outliers only on the right-hand side were checked. An estimated 95% boundary of beta (0.3, 7.2) or 19.0% was used as a cutoff interval for control outliers (Figure 8). The numbers were rounded to one decimal digit, which may have resulted in a slightly different cutoff value.

When more than three values per plate (out of six) were outside of the acceptable region, it was recommended to discard results of the entire experimental run (both plates). Using a tailored algorithm to handle experimental results (Table 2 and Table 3), only one out of 15 runs was qualified as having faulty controls.

### 3.4. Detecting Outliers in the Sample Observations

Each level of the sample dilution was repeated four times (twice per plate). To detect potential experimental errors, the Dixon’s Q test was used as a statistical method to quickly detect gross errors in small samples [23]. To detect outliers, an algorithm was used to conduct Dixon’s Q test (Table 4). Here, the critical value of 0.829 was used for a 95% confidence level at N = 4, which is rather conservative. In this cohort of plates, less than 2% of the sample observations were outliers. A tailored algorithm was used for end-to-end sample dilution results processing (Table 5).

### 3.5. Assay Variability

In order to evaluate interassay variability, a high-titer product was tested 20 times with specific NT_50_ values calculated. This testing demonstrated an average NT_50_ of 252 with a standard deviation of 74 (Table 6).

### 3.6. Specificity

In order to determine the specificity of the FRNA relative to qualitative total IgG ELISAs, a small panel of randomly selected samples of pre-determined titers were tested in the FRNA and two commercially available ELISAs. These comparisons found good agreement between the FRNA NT_50_ and ELISA positivity (Table 7). However, there was variability in some samples that were weakly positive by FRNA NT_50_. This variability was likely driven, in part, by the relatively high lower limit of detection (1:40 dilution) used in the FRNA screening assay. If less dilute samples were tested (e.g., 1:10 or 1:20), the FRNA would probably identify borderline positive samples.

## 4. Conclusions

The FRNA is a specific and highly rigorous evaluation for the presence of neutralizing antibodies in a test sample. Unlike typical neutralization assays, this assay does not rely on the subjective determination of cell cytopathic effects nor the development of multicellular plaques or immunofoci using a low number of infectious particles per well. This assay quantifies individual cells that are infected with the initial addition of virus and does not require multiple rounds of viral replication. Extensive propagation of virus could confound results due to the presence of neutralization escape variants in a viral population and secondary infection of neighboring cells leading to the formation of plaques or foci. The assay described here is also highly quantifiable, with over 4000 cells counted per individual sample dilution tested, compared to several hundred or fewer quantified plaques or foci in other assay types. While this assay is highly quantitative, it also relies heavily on high-content imaging instrumentation, which may not be available in many laboratory settings.

The inclusion of statistical evaluation of primary data provides additional rigor and is only possible due to the large number of quantifiable events recorded in the assay. The ability to “pass” or “fail” wells or test plates if data fall outside pre-established acceptance criteria provides confidence that determined titers are accurate. The use of 12-step dilution schemes and four-parameter logistical analysis to quantify a specific NT_50_, rather than reporting data as the reciprocal dilution value, provides a more precise understanding of neutralization capacity, particularly when evaluating monoclonal antibodies or nanobodies. The calculation of NT_50_ values also allows more accurate extrapolation of NT_80_ or NT_90_ values based on the calculated NT_50_ and the slope of the curve determined from test values.

The use of neutralization data has been critical to identifying potential convalescent plasma donors for hyper-immune intravenous immunoglobulin (IVIg) clinical trials. These trials are evaluating the potential therapeutic benefit of IVIg for treatment of SARS-CoV-2 infection in both hospitalized and ambulatory patients. Further, considerable effort by the World Health Organization (WHO) and others has focused on correlating data from a range of SARS-CoV-2-specific ELISAs to neutralization assays with the objective of understanding both the diagnostic and predictive value of these and other point-of-care diagnostic tools [24,25,26,27,28]. Although neutralization assays, including the one described here, are not typically used as diagnostic tools, we showed that our assay correlates well with a commonly used diagnostic ELISA. Due to requirements to handle wild-type SARS-CoV-2 in a BSL-3 facility, many laboratories have developed pseudotype-virus neutralization assays using lentivirus, vesicular stomatitis virus, and other platforms [29]. The advantage of the pseudotype systems is that they can be used at BSL-2 using reporter genes, such as luciferase or a fluorescent protein (e.g., NeonGreen) [10,29,30]. Disadvantages of such systems are that they generally only contain the SARS-CoV-2 spike protein and that the arrangement and organization of the virus spike on the pseudotyped-virus surface is unlikely to be representative of wild-type SARS-CoV-2. In many cases, such as testing plasma or a polyclonal antibody, the presentation of the spike protein is less of a concern given the array of antibodies. However, when testing monoclonal antibodies, a pseudotype-virus system could provide very different results relative to tests with wild-type virus, particularly if stoichiometric effects or cross-linking between spike proteins is a mechanism used in blocking virus attachment to the ACE2 receptor or in blocking spike trimer rearrangement as a part of the virus fusion process. Thus, any monoclonal antibody screened in a pseudotype virus assay should be validated in a live, wild-type virus assay.

Here, we describe a semi-high throughput highly quantitative neutralization assay for SARS-CoV-2 built around the Operetta high-content imaging system. While the Operetta is the preferred platform in our facility, similar equipment could be equally effective if appropriately validated. If high content imaging systems are not available, this assay could be run as an immunofocus assay, but much of the power and throughput of the assay would be lost. While there will always be variability in live-virus assays due to the nature of the biological system, using a rigorous statistical approach to inform acceptance of data can mitigate the potential negative effects of poor infection efficiency, pipetting errors, edge effects, and inconsistent staining.

## Figures and Tables

**Figure 1 viruses-13-00893-f001:**
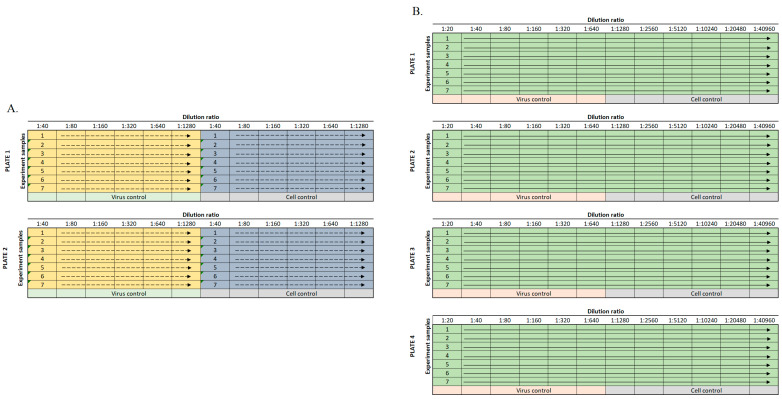
Sample dilution plate map for standard assay. (**A**). Up to seven samples were diluted twice for six dilution points on each plate and run on two duplicate plates. The yellow and blue shading indicate different replicates of each sample tested. (**B**). If more dilution points were required, samples were diluted across the entire plate for 12 dilution points and subsequently run on four plates.

**Figure 2 viruses-13-00893-f002:**
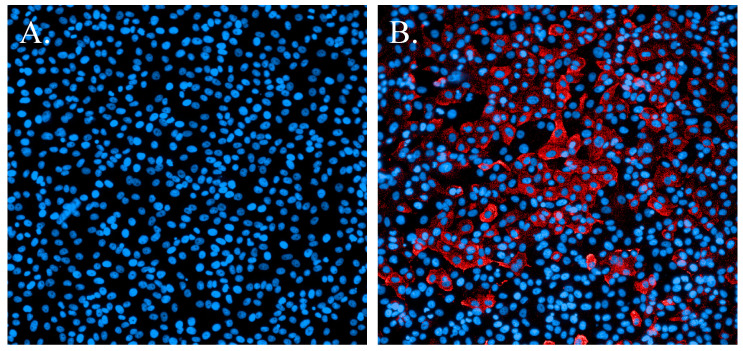
Immunofluorescence staining of SARS-CoV-2-infected cells. (**A**). Non-infected cells stained with Hoechst nuclear stain (blue). (**B**). Cells infected with SARS-CoV-2 and probed with a SARS-CoV N-protein-specific antibody and Alexa594 secondary antibody (red). Cells were counterstained with Hoechst nuclear stain (blue).

**Figure 3 viruses-13-00893-f003:**
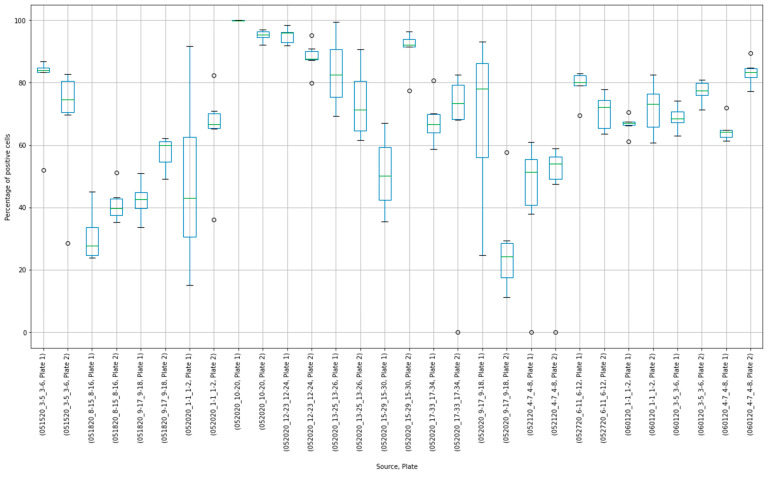
Box plot results of virus control observations.

**Figure 4 viruses-13-00893-f004:**
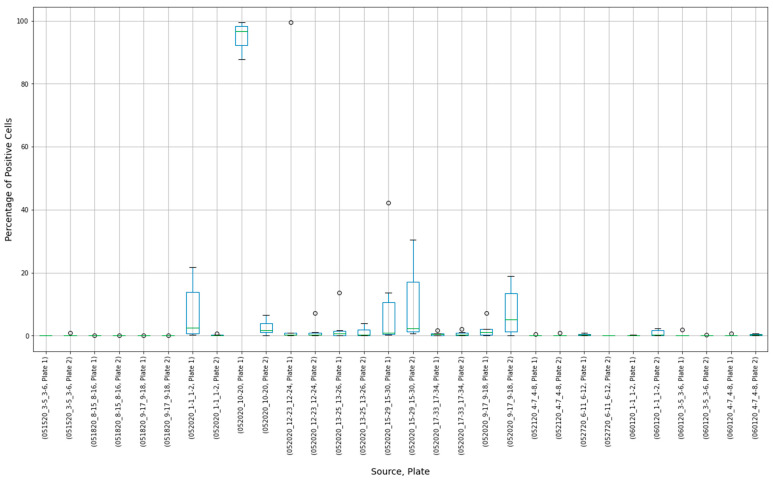
Box plot results of cell control observations showing variability between controls in individual experimental runs.

**Figure 5 viruses-13-00893-f005:**
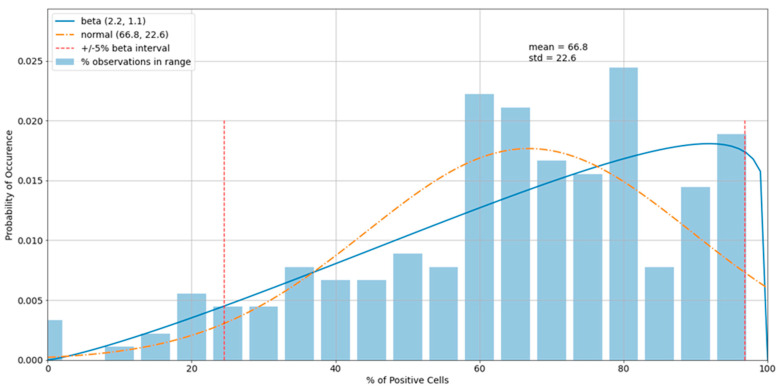
A histogram of the variability of virus control observations with fitted normal and beta distributions.

**Figure 6 viruses-13-00893-f006:**
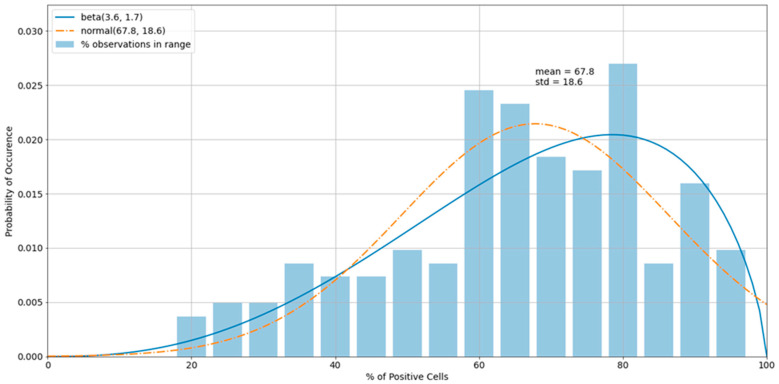
A histogram of the virus control observations (Figure 5) after outliers were removed.

**Figure 7 viruses-13-00893-f007:**
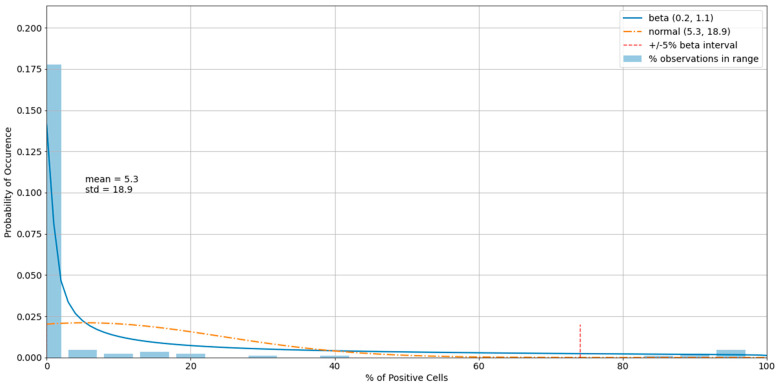
A histogram of the variability of cell control observations with fitted normal and beta distributions.

**Figure 8 viruses-13-00893-f008:**
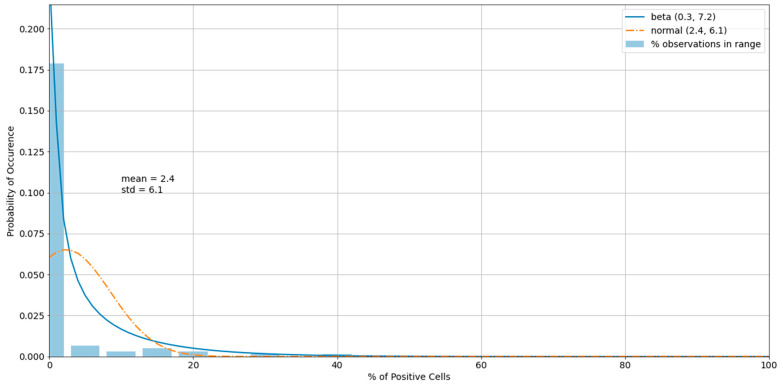
A histogram of the cell control observations (Figure 7) after outliers were removed.

**Table 1 viruses-13-00893-t001:** Assay setup parameters.

Parameter	Value
Cell seeding density	30,000 cells per well
Dulbecco’s Modified Eagle Medium without calcium	
Virus multiplicity of infection	0.5
Virus/sample neutralization period in dilution block	1 h, 37 °C, 5% CO_2_
Virus/sample incubation with permissive cells	24 h, 37 °C, 5% CO_2_

**Table 2 viruses-13-00893-t002:** Algorithm of handling experimental results for the virus control.

Step	Purpose	Actions
1	Mask virus control outliers.	Exclude values outside of the critical region (<5%, >95%) of the beta distribution estimated for virus control observations.
2	Quality check the plates.	If the number of non-masked values of per plate is less than 3, then discard the results of entire experiment. Otherwise, go to Step 3.
3	Calculate the mean of virus control.	Use non-masked values from both plates to calculate the mean of virus control.
4	Calculate FRNA_50_.	Divide the mean of virus control by 2.

FRNA_50_ = fluorescence reduction neutralization assay at 50% reduction.

**Table 3 viruses-13-00893-t003:** Algorithm of handling experimental results for the cell control.

Step	Purpose	Actions
1	Mask cell control outliers.	Exclude values outside of the critical region (>95%) of the beta distribution estimated for cell control observations.
2	Quality check of the plates.	If the number of non-masked values of per plate is less than 3, then discard the results of the entire experiment. Otherwise, go to Step 3.
3	Calculate the mean of cell control.	Use non-masked values from both plates to calculate the mean of cell control.
4	Calculate FRNA_100_.	Use the mean of cell control.

FRNA_100_ = fluorescence reduction neutralization assay at 100% reduction.

**Table 4 viruses-13-00893-t004:** Algorithm for conducting Dixon’s Q test to detect sample outliers.

Step	Purpose	Actions
1	Obtain the maximum value, Q_max_.	Obtain the difference between the maximum of four observations and the second largest value. Divide it by the range between the maximum and the minimum.
2	Obtain the minimum value, Q_min_.	Obtain the difference between the second smallest value and the minimum of four observations. Divide it by the range between the maximum and the minimum.
3	Compare with Q_95_ at 95% confidence level.	Q_95_ is 0.829. If Q_max_ or Q_min_ is above Q_95_, then mask that observation. If both are masked, discard the sample.

**Table 5 viruses-13-00893-t005:** Algorithm of handling sample dilution results.

Step	Purpose	Actions
1	Check controls.	If at least one plate from virus control or cell control fails, discard the results. Otherwise, go to Step 2.
2	For each dilution ratio, check four observations of the sample.	Use Dixon’s Q test to check whether the minimum and maximum values of the sample are outliers. If both are rejected, then discard the results. If one is rejected, then remove it from calculations and go to Step 3.
3	Calculate the means.	Use non-masked values to calculate the mean of each dilution ratio.
4	Compare with FRNA thresholds.	Compare dilution means with FRNA_50_ and FRNA_100_.

FRNA = fluorescence reduction neutralization assay. FRNA_50_ = fluorescence reduction neutralization assay at 50% reduction. FRNA_100_ = fluorescence reduction neutralization assay at 100% reduction.

**Table 6 viruses-13-00893-t006:** FRNA experimental variability testing.

Run	NT_50_		Run	NT_50_
1	173		11	335
2	267		12	262
3	230		13	359
4	274		14	219
5	337		15	205
6	300		16	231
7	257		17	147
8	258		18	292
9	403		19	191
10	108		20	185
			Mean:	252
			Standard Deviation:	74

NT_50_ = 50% neutralizing titer.

**Table 7 viruses-13-00893-t007:** Comparison of ELISA data to the FRNA NT_50_.

Sample	FRNA NT_50_		R&D ELISA SARS-CoV-2			EURO ELISA SARS-CoV-2		
	Rep 1	Rep 2	Rep 1	Rep 2	Rep 3	Rep 1	Rep 2	Rep 3
1	<40	80	+	+	+	+	+	+
2 *	40	40	+	-	-	+	+	+
3	<40	<40	-	+	+	-	-	-
4	<40	<40	+	+	+	+	+	+
5	<40	<40	-	-	-	-	-	-
6	<40	<40	+	+	+	+	+	+
7	160	80	+	+	+	+	+	+
8	80	40	+	+	+	+	+	+
9	640	320	+	+	+	+	+	+
10	80	80	+	+	+	+	+	+
11	160	80	+	+	+	+	+	+
12	320	320	+	+	+	+	+	+
13	<40	<40	-	-	-	-	-	-
14	<40	<40	-	-	-	-	-	-
15	80	<40	+	+	+	+	+	+
16	<40	<40	-	-	-	-	-	-
17	<40	<40	-	-	-	-	-	-
18	40	<40	+	+	+	+	+	+
19	320	40	+	+	+	+	+	+
20	<40	40	-	-	-	-	-	-
21	80	<40	+	+	+	+	+	+
22	80	80	+	+	+	+	+	+
23	<40	<40	-	-	-	-	-	-
24	40	<40	+	+	+	+	+	+
25	<40	<40	-	-	-	-	-	-
26	<40	<40	-	-	-	-	-	-
27	<40	<40	+	+	+	+	+	+
28	80	<40	+	+	+	+	+	+
29	80	80	+	+	+	+	+	+
30	<40	<40	+	+	+	+	+	+
Positive Control	80	80	+	+	+	+	+	+

* Gray shaded rows indicate samples with variability between ELISA and NT_50_ results. FRNA = fluorescence reduction neutralization assay. NT_50_ = 50% neutralizing titer. ELISA = Enzyme-Linked ImmunoSorbent Assay. R&D = R&D Systems. EURO = EuroImmun.

## Data Availability

Data associated with this study are included in the manuscript. Additional primary data can be provided by request.

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
