# Peer review of "Scalable, Micro-Neutralization Assay for Assessment of SARS-CoV-2 (COVID-19) Virus-Neutralizing Antibodies in Human Clinical Samples"

_viruses, 2021, doi:10.3390/v13050893_

Round 1

Reviewer 1 Report

The manuscript by Bennett et al describes the optimization of the FRNA assay for the qualitative assessment of SARS-CoV-2 neutralizing antibody titers. The assay was initially developed for MERS and only slightly modified for the SARS-CoV-2. While the manuscript is very well written, and data is clearly presented a few concerns arise regarding the robustness and reproducibility of the assay. First, the positive control values seem to be unequally distributed over a wide range of values, far from the gaussian distribution. It would be helpful if the authors could comment on why these issues arise (technical issues in method or else) and if this was also observed for other neutralization assays (TCID50, plaque assay, etc). In addition, the neutralization values obtained by this assay should be compared with the values provided by the current ‘gold standard’ (qPCR or plaque assay). Another issue that should be addressed is the applicability of the assay when using only a small number of samples/plates, as a large number of positive control repeats is required for the rigorous statistical analysis employed here.

Minor question:

What concentration of anti-rabbit IgG was used?

Author Response

We observe a certain percent of values that fall outside critical boundaries of theoretical distributions, including the gaussian distribution. Presence of extreme values, or outliers, can greatly affect estimated values of NT50. The statistical analysis shown in Figures 4 – 8, demonstrates the impact of the outliers on the estimates of the mean and standard deviation of the percent of positive cell values.

 In this article, we could not address the issues of exactly why outlier values show up in the data (contamination of control samples, technical errors, cell passage number, etc.). More rigorous research would be required to identify root causes of the errors in order to refine FRNA procedure, although some variability is inescapable since this is a biological system. A larger number of control observations will help obtain a better measurement of naturally occurring outliers for this procedure. Future work can be devoted to determine the fraction of Type I (removing a valid observation from the control sample) and Type II (keeping the “bad” value in the control sample) errors due to these technical factors.  

 The proposed algorithm, Dixon’s Q test, to detect and remove outliers, allows automatically correct estimates of the FRNA50 and FRNA100 and reduces time that researchers spend on processing test results.   

Other than the ELISA comparisons, we have not directly compared this assay to others.  However, this assay has been compared against other assays (e.g. plaque assay, TCID50, pseudovirus, etc.) by collaborators using standardized panels of serum or plasma. Unfortunately, these data are not published yet.  Comparing to a PCR assay would not be helpful since the assays are measuring two very different things and our assay is not meant to be a diagnostic tool.

I am not sure I understand the point of the last sentence.  We run positive controls with each set of plates and use those for comparison in each run.  The algorithms have already been developed and are applied to data from each run.

Minor question:

What concentration of anti-rabbit IgG was used?

This information was added to the manuscript, line 134

Reviewer 2 Report

In this study, the authors utilized a previously developed fluorescence reduction neutralization assay (FRNA) and adapted its use for the detection of SARS-CoV-2 specific neutralising antibodies. In general, this is a well-written study that adds to the increasing toolbox of SARS-CoV-2 specific assays. However, there are some concerns. General concern: The assay described here requires high-content imaging instrumentation, rendering it unsuitable for many laboratories, questioning its general implementability. Specific concerns: Regarding specificity, the authors found discrepancies between their assay and two commercially available ELISAs, especially at lower NT50 concentrations. Table 7 is confusing. Why are two values depicted for the FRNA NT50 and 3 columns for the respective ELISAs? How does their assay compare to currently used standard assays for neutralisation? A direct comparison would be helpful to judge this assays performance. Can similar NT50 titers be calculated? Minor point: Please add more information in the figure legend. It is not clear what exactly is being depicted in the figures and tables. 

Author Response

We fully recognize that this assay is not one that could be run in all laboratories given the equipment requirements.  The point was to show what can be done if high-content imaging systems are available and to demonstrate the approach we use to improve the rigor of the assay.  We did modify the discussion to suggest a potential option if high-content imaging is not available in a specific lab (line 373: If high content imaging systems are not available, this assay could be run as an immunofocus assay, but much of the power and throughput of the assay would be lost.)

In table 7 the two columns for the NT50 are two independent repeats of the same samples.  We ran three repeats for the two ELISAs that were tested, hence three columns for each.  We modified the table somewhat to make it clear that each column is a replicate assay run.

This assay has been compared to other live virus and pseudovirus assays and is similar but less sensitive to others.  Based on current operational parameters, there is no ‘standard’ neutralization assay for SARS-CoV-2.  However, there are international (WHO) and national (US) standards that have been developed and we are currently assessing our assay with these test materials so it can be compared to other assays.

The information explaining the data in the figures is in the text.  Adding the same information to the legends would be redundant and probably confusing.

Reviewer 3 Report

The manuscript by Bennett et al., was a relatively comprehensive research article on the potential roles of neutralizing antibodies and viral presence for the SARS-CoV-2 infected samples. The authors focused on the scientific issues for the use of passive immunization and immunoglobin therapy and tested the antibody-based assays of SARS-CoV-2. The authors compared fluorescence-based readouts and viral protein ELISA to detect the clinical samples with COVID-19. The experiments were well performed and data was well collected. The whole study still generated a concern how specific patients with clinical diagnosis with COVID-19 or SARS-CoV-2 infection. The results and discussion (as shown in Lines 324-325), like “the presence of neutralizing antibodies in a test sample” might not be comparable based on the study. The reason was that, the neutralizing antibody could be IgA, IgM in stead of IgG; the antibody assay was also based on artificial IgG and viral protein. There were missing statistical evidence for the conclusive words.

Author Response

The question in the reviewer’s comment is not clear.  The assay described in the submitted manuscript tests for neutralizing antibodies and does not discriminate between antibody isotypes and is not a diagnostic assay due to it’s reduced sensitivity relative to ELISAs.  We have tested material from acutely ill individuals who have a very high neutralizing titer that is likely the result of high IgM titers.  It also isn’t clear what statistical evidence might be missing as indicated by the reviewer.

Round 2

Reviewer 2 Report

The authors have addressed my comments sufficiently.

Author Response

Thank you

Reviewer 3 Report

The manuscript by Bennett et al. was improved but there were still a few minor concerns to be addressed:

  • The qualitative assessment (such as in the title) was not properly summarized.
  • It would be better to provide photos/pictures for micro-neutralization assay.
  • Cell source was not well justified in the revised manuscript.
  • How different it was between the SARS-CoV-2 S1 protein-based detection and N protein-based measurement was not clearly discussed.
  • The proposed “highly quantitative neutralization assay” (Line 370) was mentioned; however, the authors did not provide any data linking FRNA with the viral load or antibody titer, which significantly affected the quality of the research.

Author Response

We would like to thank the reviewer for comments focused on improving our submission.  We have done our best to address comments directly and have made modifications to the manuscript in an effort to be more clear.

  • The qualitative assessment (such as in the title) was not properly summarized.
    • We agree that the use of the word “Qualitative” in the title is inappropriate since the assay described here carefully quantifies the titer of SARS-CoV-2 neutralizing antibodies in plasma and serum.
  • It would be better to provide photos/pictures for micro-neutralization assay.
    • Initially we had an example image of an antibody dilution series, but we opted to remove the image since we felt it really didn’t add much to the manuscript other than a pretty picture.  Figure 2 gives an example of infected vs uninfected cells.  The dilution series image is really just a series of similar pictures.
  • Cell source was not well justified in the revised manuscript.
    • The point of this question isn’t clear.  The source of both the Vero and Vero E6 cells is provided in the methods section.  Vero cells are the preferred cell line for this type of assay as they are easy to work with and can be infected.  Over the course of the past year, many labs have tested various versions of Vero cells (e.g. Vero-ACE2, Vero-SLAM, Vero-TMPRSS).  Since we have been running this assay for over a year and with a specific purpose, we have not tested other cell lines.  However, the assay is adaptable, only needing optimization of MOI and incubation time for the specific virus being tested.
  • How different it was between the SARS-CoV-2 S1 protein-based detection and N protein-based measurement was not clearly discussed.
    • It isn’t clear what is being asked here.  Assuming this is in reference to our comparison between a Spike-based ELISA assay and our use of an N specific detection antibody, we added text to the discussion (lines 359-361) indicating we found good correlation between our assay and the commercially available ELISA we tested.
  • The proposed “highly quantitative neutralization assay” (Line 370) was mentioned; however, the authors did not provide any data linking FRNA with the viral load or antibody titer, which significantly affected the quality of the research.
    • A comparison between viral load and our FRNA titer is not necessary since we know exactly how much virus we put into the system (MOI) and the number of infected cells (target ~50%).  The actual virus present when we run the assay is not relevant as long as there are a sufficient number of infected cells.  In regards to the question of antibody titer, that is what the assay measures.  However, rather than measuring total antibody like an ELISA, this assay measures functional (neutralizing) antibody.